# Plastid Genomes of the Early Vascular Plant Genus *Selaginella* Have Unusual Direct Repeat Structures and Drastically Reduced Gene Numbers

**DOI:** 10.3390/ijms22020641

**Published:** 2021-01-11

**Authors:** Hyeonah Shim, Hyeon Ju Lee, Junki Lee, Hyun-Oh Lee, Jong-Hwa Kim, Tae-Jin Yang, Nam-Soo Kim

**Affiliations:** 1Department of Agriculture, Forestry and Bioresources, Plant Genomics & Breeding Institute, Research Institute of Agriculture and Life Sciences, College of Agriculture & Life Sciences, Seoul National University, 1 Gwanak-ro, Gwanak-gu, Seoul 08826, Korea; ehs1681@snu.ac.kr (H.S.); imjoo126@naver.com (H.J.L.); archejun@gmail.com (J.L.); dlgusdh88@naver.com (H.-O.L.); 2Phyzen Genomics Institute, Seongnam 13558, Korea; 3Department of Horticulture, Kangwon National University, Chuncheon 24341, Korea; jonghwa@kangwon.ac.kr; 4Department of Molecular Bioscience, Kangwon National University, Chuncheon 24341, Korea

**Keywords:** *Selaginella*, lycophytes, plastomes, direct repeats, RNA editing

## Abstract

The early vascular plants in the genus *Selaginella*, which is the sole genus of the Selaginellaceae family, have an important place in evolutionary history, along with ferns, as such plants are valuable resources for deciphering plant evolution. In this study, we sequenced and assembled the plastid genome (plastome) sequences of two *Selaginella tamariscina* individuals, as well as *Selaginella stauntoniana* and *Selaginella involvens*. Unlike the inverted repeat (IR) structures typically found in plant plastomes, *Selaginella* species had direct repeat (DR) structures, which were confirmed by Oxford Nanopore long-read sequence assembly. Comparative analyses of 19 lycophytes, including two *Huperzia* and one *Isoetes* species, revealed unique phylogenetic relationships between *Selaginella* species and related lycophytes, reflected by structural rearrangements involving two rounds of large inversions that resulted in dynamic changes between IR and DR blocks in the plastome sequence. Furthermore, we present other uncommon characteristics, including a small genome size, drastic reductions in gene and intron numbers, a high GC content, and extensive RNA editing. Although the 16 *Selaginella* species examined may not fully represent the genus, our findings suggest that *Selaginella* plastomes have undergone unique evolutionary events yielding genomic features unparalleled in other lycophytes, ferns, or seed plants.

## 1. Introduction

Chloroplasts, representing the most typical form of plastids, are semiautonomous cellular organelles found in photosynthetic plants and algae that contain their own genomes. Plastid genomes (plastomes) are typically 120–160 kb long, with a quadripartite architecture comprising one long single-copy (LSC) region and a short single-copy (SSC) region separated by two inverted repeats (IR_A_ and IR_B_) [1]. Plastomes contain approximately 120 genes, in which most encoding proteins function in photosynthesis, protein synthesis, and DNA replication [2,3]. Although the gene order and genome architecture have been broadly preserved across taxa, plastomes have undergone remarkable genome reduction and rearrangements over the course of plant evolution. Classic examples include the almost complete loss of IRs in conifers [4,5], IR expansion and contraction in some monilophyte ferns [6], several inversions and the loss of IR regions in some legumes [7] and ferns [8], and the loss of most or all *ndh* genes in distantly related fern lineages [9] and in seed plants including both gymnosperms [10,11] and angiosperms [12,13,14,15].

Pteridophytes are free-sporing vascular plants comprising two classes—Lycopodiopsida (lycophytes) and Polypodiopsida (ferns)—which form distinct evolutionary lineages in the tracheophyte phylogenetic tree [16]. Lycopodiopsida is an ancient lineage that diverged shortly after land plants evolved to acquire vascular tissues [17]. Although lycophytes were abundant and dominant in land flora during the Carboniferous era [18], only three orders are currently recognized within Lycopodiopsida, including Lycopodiales, Isoëtales, and Selaginellales [16]. Selaginellales contains the single family Selaginellaceae, which consists of the single genus *Selaginella* [19,20,21]. The *Selaginella* genus contains over 700 species distributed in a diverse range of habitats, including deserts, tropical rain forests, and alpine and arctic regions.

Analyses of plastome sequences have revealed that considerable genomic changes have occurred in some lineages of pteridophyte species. Comparative analyses with moss plastomes revealed five inversions in the fern plastomes [8]. One ~3.3 kb inversion in the LSC region existed in all ferns [22]. A pair of partially overlapping inversions was mapped to the IR region in the common ancestor of most fern species and a second pair of overlapping inversions was found in core leptosporangiate ferns [23]. Gene losses were also prominent events in the plastomes in some lineages, including the loss of *chlB*, *chlL*, and *chlN* in *Psilotum* SW and *Tmesipteris* [6,24] and the complete loss of *ndh* genes and reduction of the small single-copy length in Schizaeceae [9]. Therefore, comparing the components of plastomes with those of whole genomes or proteome sequences provides important insights into plant phylogeny and evolution [9,22,25,26].

Since the release of the first plastome sequence for the lycophyte *Huperzia lucidula* [27], complete plastome sequences have been made available for various *Huperzia* species (Lycopodiales) [28,29], *Isoetes flaccida* (Isoëtales) [22], and *Selaginella* species (Selaginellales) [30,31,32,33,34,35,36]. A comparative analysis of the plastomes of *H. lucidula* with those of bryophytes and seed plants revealed a 30 kb inversion in *H. lucidula* and seed plants [27]. Karol et al. compared the plastome structures of mosses, a hornwort (*Anthoceros formosae*), and four lycophytes (*H. lucidula*, *I. flaccida*, *Selaginella moellendorffii*, and *Selaginella uncinata*) and identified an inversion between the plastomes of *I. flaccida* and *Selaginella* species and a microinversion between *H. lucidula* and *I. flaccida*. Furthermore, the gene *ycf2* is present in *Huperzia*, but has been deleted in *Isoetes* [22]. Tsuji et al. showed that the gene order and arrangement are almost identical between the plastomes of *H. lucidula* and bryophytes, but the plastome of *S. uncinata* is considerably different from those of bryophytes, which were derived from a unique inversion event, transpositions, and many gene losses [30]. 

One notable feature of *Selaginella* plastomes is the direction of the repeat blocks. Although the near or complete loss of IRs has been reported in some plant lineages of conifers and Fabaceae [4,5], the two repeats are inversely oriented as IR_A_ and IR_B_ in most plastomes [37,38]. Recent studies, including the current report, however, have revealed that the repeat blocks are direct repeats (DR_A_ and DR_B_) in most plastomes in the genus *Selaginella* [33,34,35,36]. Other distinct features of the plastomes of *Selaginella* species are a genus-wide GC bias and the high occurrence of RNA editing. An analysis of the 3507 plastome sequences from algae to seed plants at the National Center for Biotechnology Information (NCBI) as of November 2019 [39] revealed an average GC content of 37.38 ± 2.26%. However, the average GC content of the five *Selaginella* plastomes was 52.8%, ranging from 51.0% in *S. moellendorffii* to 54.8% in *S. uncinata* [34]. RNA editing is a prominent feature of organelle genes, including *Selaginella* plastid genes. Whereas about 200–500 sites of C-to-U RNA editing have been detected in flowering plant plastomes, more than 3400 C-to-U RNA editing events were discovered in *S. uncinata* plastomes [32]. This extensive RNA editing is thought to be related to the high GC bias due to a combination of the reduced AT-mutation pressure and the high number of C-to-U RNA editing sites in the *Selaginella* genus [31].

In the current study, we sequenced the plastomes of three *Selaginella* species—*Selaginella tamariscina*, *Selaginella stauntoniana*, and *Selaginella involvens*—via whole-genome sequencing (WGS) using both the Illumina sequencing platform and the Oxford Nanopore long-read platform and assembled their complete plastomes. We performed a genus-wide comparison of genomic features, including GC contents, structural changes in the genome, and gene losses. This analysis uncovered remarkably dynamic features of the plastomes of plants of the *Selaginella* genus in the Division Lycopodiophyta.

## 2. Results

### 2.1. Selaginella Plastomes Contain Reduced LSCs and Unusual DR

In the current study, we sequenced, assembled, and annotated the plastomes of three *Selaginella* species, including *S. tamariscina, S. stauntoniana,* and *S. involvens* (Figure 1). We analyzed these assembled plastome sequences, along with the sequences of 13 other species in the *Selaginella* genus and three species from other orders (Isoetales and Lycopodiales) in the Lycopodiopsida, which were obtained from NCBI (Table 1). The plastome sizes of *Selaginella* species ranged from 110,411 to 147,148 bp, with an average of 132,571 ± 11,612, while the plastomes of non-*Selaginella* lycophytes ranged from 145,303 to 154,373 bp. Overall, *Selaginella* species had reduced LSC regions, with some species having LSCs that were almost half the size of those of non-*Selaginella* lycophytes. Most *Selaginella* species had longer SSCs than LSCs, except for the three species of *Selaginella lepidophylla, Selaginella hainanensis,* and *S. uncinata* (Table 1).

Most *Selaginella* species shared a unique plastome structure consisting of a set of direct repeats (DRs) instead of the inverted repeats (IRs) found in most plastomes. This DR structure was confirmed by PCR amplification using primer combinations designed based on the junction sites for the hypothetical DR and IR structures (Figure 2a). The primer combinations only differed in the sequence of the first repeat block (the inverted form of DR_B_ in Figure 2a). PCR amplification and gel electrophoresis showed that only primer combinations for the direct repeat structure were amplified, suggesting that the direct repeat structure was present in the plastome (Figure 2b). We confirmed the unusual DR structure through the assembly of the *S. tamariscina* plastome using Oxford Nanopore Sequencing Technology and an assembly pipeline we established. The plastome was thereby assembled into two contigs of 61,029 and 71,944 bp in size (Figure 2c). These contigs covered the junction sites between the single-copy regions and the DR regions without inversions, confirming the direct repeat structure of the *S. tamariscina* plastome.

### 2.2. Selaginella Plastomes Contain Fewer Genes than Other Lycophyte Plastomes

The number of plastome genes differed significantly among lycophyte species, pointing to the frequent occurrence of gene loss events. In particular, the plastomes of *Selaginella* species generally had fewer protein-coding and transfer RNA (tRNA) genes than those of non-*Selaginella* species (Table 1). Within the genus, *S. hainanensis* had the highest number of plastome genes (103 genes), whereas *S. tamariscina* had the fewest (75 genes). *Huperzia* and *Isoetes* species contained almost all plastid genes. All 19 lycophyte species analyzed carried two copies of the four ribosomal RNA (rRNA) genes located in the repeat regions.

Figure 3 shows gene polymorphisms (presence/absence/pseudogenes) among the plastomes of the 19 lycophyte species. *Isoetes* and *Huperzia* species contained most plastid genes, except for a few that were lost or pseudogenized. In *Selaginella* species, many genes were absent or pseudogenized, specifically tRNA genes, ribosomal protein genes, and *ndh* genes. While non-*Selaginella* species had an average of 29.7 tRNA genes, *Selaginella* species averaged 11.6 tRNA genes, i.e., less than half the number in non-*Selaginella* species. The *S. stauntoniana* and *S. tamariscina* plastomes each contained only six tRNA genes. There were also many losses of protein-coding genes, such as the loss or pseudogenization of most *ndh* genes in *S. lepidophylla*, *Selaginella lyallii*, *Selaginella indica*, *S. stauntoniana*, *S. tamariscina*, *Selaginella vardei*, and *Selaginella bisulcata*, as well as losses of many ribosomal subunit genes throughout the genus. Other genes lost in *Selaginella* species included genes related to lipid biosynthesis (*accD*), translation initiation (*infA*), and other miscellaneous functions (*ycf4*, *ycf66*, *cemA*, and *matK*).

### 2.3. Phylogenetic Analysis Shows Two Main Lineages with Dynamic Structural Variations

Phylogenetic analysis using the BEAST software separated the 19 lycophyte species into three major groups, including one group for non-*Selaginella* species and two lineages within the *Selaginella* genus, all divided by two main divergence events (Figure 3). The first group (labeled 1 in Figure 3) consists of *H. lucidula*, *H. serrata*, and *I. flaccida*, which contain IRs. The *Selaginella* species were divided into two lineages. One lineage (labeled 2 in Figure 3) was divided into two subgroups, with one group containing *S. lyallii*, *Selaginella kraussiana*, and *Selaginella remotifolia* and the other containing *S. indica*, *S. vardei*, and *S. lepidophylla*. *S. lepidophylla* contains IRs, while the five other species contain DRs. Another lineage (labeled 2′ in Figure 3) contains 10 *Selaginella* species without apparent subgrouping. Among these 10 species, eight contain DRs, whereas *S. hainanensis* and *S. uncinata* contain IRs. 

As a result of the plastome structure comparison of all 19 species, species within the same lineages were shown to share similar structures (Figure 4). In the first group, comprising non-*Selaginella* species, the plastomes had typical plastome structures, containing IRs. *Selaginella* species exhibited a different structure by a block inversion spanning from *trnF* in the LSC to *trnN* at the junction between IR_B_ and the SSC. This inversion caused part of the LSC to be repositioned in the SSC region, resulting in the shortening of the LSC and the expansion of the SSC seen in *Selaginella* species. The block that took part in this inversion event contained one of the repeat regions, IR_B_, which ended up in the same orientation as the unaffected repeat region, thus creating a set of DRs in *Selaginella* species. The second inversion event occurred in both lineages independently (red diamonds in Figure 3). Because DR_B_ was involved in these second inversions, DR_B_ was converted back to IR_B_ in species that underwent the second inversion event in each lineage, meaning *S. lepidophylla* in lineage 2 and *S. hainanensis* and *S. uncinata* in lineage 2′ (Figure 3).

### 2.4. Plastome Diversity among Five S. tamariscina Collections Reflects the Geographical Diversity

We explored the intraspecific diversity within five *S. tamariscina* individuals collected from different regions: Two from China and three from Korea. The plastome size of *S. tamariscina* ranged from 126,365 to 126,700 bp. We compared the five plastome sequences by aligning them in a pairwise fashion. We examined the number of single-nucleotide polymorphisms (SNPs) and insertions/deletions (InDels) to analyze the intraspecies diversity in the *S. tamariscina* collections. The collections from regions within Korea showed relatively little variation, harboring up to 1 SNP and 15 InDels. However, a comparison of SNP and InDel counts in collections from Korea vs. China revealed much more variation. Plastomes of Korean collections showed approximately 1218 SNPs when compared to those of two Chinese collections: China2018 and China2019. Furthermore, between the two Chinese collections [33,35], 1246 SNPs and 401 InDels were detected (Table 2). Among the variations, insertions and deletions were located within two genes related to chlorophyll biosynthesis (*chlB* and *chlN*) that induced frameshifts resulting in premature stop codons. While China2019 retained all three *chl* genes, the other four collections contained the remains of *chlB* and *chlN* as pseudogenes that are likely nonfunctional. A BEAST phylogenetic tree of five *S. tamariscina* individuals with *S. stauntoniana*, which is the closest relative, as the outgroup showed that SNU2014, SNU2018, Korea2020, and China2018 diverged from each other around 90 thousand years ago, but diverged from China2019 around 900 thousand years ago (Figure A1). 

### 2.5. Selaginella Plastomes Exhibit Many Intron Losses and GC Bias

We identified 19 genes that contain introns (13 protein-coding genes and 6 tRNA genes) in the plastomes of the 19 lycophytes (Table 3). Of the protein-coding genes, two—*clpP* and *ycf3*—contained two introns and the other 17 contained a single intron between two exons. *Huperzia* and *Isoetes* species retained introns in most of these genes. However, the numbers of introns in the *Selaginella* plastomes was highly reduced due to either the deletion of intron-containing genes or the absence of introns in specific *Selaginella* species. For instance, *rps12, ycf66*, and *rps16* were not present in *Selaginella* species, and tRNA genes containing introns in other lycophytes were either absent or intron-less in *Selaginella*. The *clpP* gene in *Huperzia* and *Isoetes* species contained two introns: *clpP-1* and *clpP-2*. *clpP-1* was lost in species in lineage 2, but was retained in most species in lineage 2′, except for *S. stauntoniana* and *S. tamariscina*. *ycf3* also contained two introns: *ycf3-1* and *ycf3-2*. While *ycf3-1* was present in all lycophyte species analyzed, *ycf3-2* was not present in the species of *Selaginella* lineage 2.

Analysis of the GC contents of the plastomes revealed that the *Selaginella* species had considerably higher GC ratios than non-*Selaginella* species (Table 1). The GC contents of *Selaginella* species ranged from 50.75% for *S. lyallii* to 56.49% for *S. remotifolia*, whereas those of non-*Selaginella* species were significantly lower (ranging from 36.25% to 37.94%). There was no correlation between the orientation of the repeats and the GC content. Overall, *Selaginella* species lost more intron sequences compared to non-*Selaginella* lycophytes, and the intron pattern was shared in each lineage.

### 2.6. RNA Editing Is Commonly Found in Selaginella Plastomes

The annotation of the three plastomes was hindered by the frequent absence of authentic ATG start codons for protein-coding genes, as many genes contained ACG nucleotides instead of ATGs at their start sites. Stop codons were also missing in some genes. After a final manual curation of the gene annotations, we determined that 76.8%, 77.2%, and 53.3% of protein-coding genes in *S. tamariscina*, *S. stauntoniana*, and *S. involvens*, respectively, started with ACG instead of ATG. 

To validate our hypothesis about RNA editing in these sequences, we conducted a transcriptome analysis of the *S. tamariscina* plastome. We mapped raw RNA sequencing data from *S. tamariscina* onto the complete plastome sequence of the same species to determine whether the potential RNA editing sites were indeed edited in the mRNA sequences. We calculated the ratios of reads aligned to the start and stop codons in the unedited (C nucleotide) vs. edited (T nucleotide) forms using CLC find variations (ver. 4.3.0). In the *S. tamariscina* plastome sequence, 43 of the 56 genes contained potential editing sites for start codons, and four genes contained potential editing sites for stop codons (Figure 5). The coexistence of reads aligned to the same start and stop codons with either the edited or unedited codon indicated that these genes had undergone C-to-U RNA editing. Forty-two of the 43 potential editing sites were confirmed to have undergone C-to-U RNA editing. In contrast, there was a lower rate of potential RNA editing sites in stop codons, but three out of the four potentially edited sites were indeed edited. For start codons, most of the genes (except *atpE*, *chlL*, *psbN*, and *rpoB*) had a higher ratio of RNA-edited reads with the T nucleotide (switched from the U nucleotide during RNA sequencing) than unedited reads with the C nucleotide.

Overall, *Selaginella* species had a higher rate of abnormal start codons (ranging from 23.33% of the genes for *S. lyallii* to 77.19% in *S. stauntoniana*), while the *Huperzia* species had very low percentages (<10%), and *I. flaccida* was on the lower end of the range for *Selaginella* species (Figure 6). In regard to abnormal stop codons, *Selaginella sanguinolenta* had the highest percentage (41.79%), and the levels for most other *Selaginella* species were similar; the exceptions were *S. lyallii*, *S. kraussiana*, *S. lepidophylla*, *S. stauntoniana*, and *S. tamariscina*, with percentages < 10%. Some codons (marked as ‘others’) did not fall into either category, as they lacked authentic start or stop codons, but did not contain codons that were potential targets for RNA editing (Figure 6). A survey of start and stop codons in all 19 lycophyte plastid genes revealed a generally higher frequency of potential editing sites in start codons compared to stop codons (Figure 6). The percentages of potential RNA editing sites in start codons were 29.27%, 8.14%, and 4.71% for *I. flaccida*, *H. lucidula*, and *H. serrata*, respectively, and among the *Selaginella*, they ranged from 23.33% for *S. lyallii* to 77.19% for *S. stauntoniana*. Within the genus, *S. stauntoniana* and *S. tamariscina* had the highest frequency of potential RNA editing sites (77.19% and 76.79%, respectively). In stop codons, the proportions of potential RNA editing sites ranged from 4.71% to 5.81% for *H. serrata* and *H. lucidula*, respectively, while those for *Selaginella* species ranged from 0% for *S. kraussiana* to 41.79% for *S. sanguinolenta* (Figure 6). 

## 3. Discussion

### 3.1. Unique Features of Selaginella Plastomes

Overall, our results support previous findings that the plastome sequences of *Selaginella* species were smaller than those of non-*Selaginella* and typical land plants [34]. As of February 2019, 2364 complete plastome sequences of land plants were registered in NCBI, with an average size of 151,167 ± 34,672 bp. In contrast, the average size of the 16 *Selaginella* plastomes we assessed was 132,571 ± 11,612 bp. Although the canonical quadripartite plastid genomic structure was retained in these *Selaginella* species, directional change occurred in most *Selaginella* species with SSC expansion and LSC contraction. The relative length of the SSC as a proportion of the total plastome ranged from 24.96% (*S. moellendorffii*) to 42.65% (*S. involvens*), averaging 35.73%. These values are higher than those in non-*Selaginella* species (12.73–18.72%) (Table 1), as well as those in eight selected monilophyte ferns (ranging from 11.7% in *Psilotum nudum* to 15.57% in *Adiantum capillus-veneris*, averaging 13.84%) [6]. The expansion of the SSC region in *Selaginella* species contrasts with the SSC contraction in Schizaeceae ferns, in which the SSC was reduced by as much as 2255 bp to contain only two genes in *Schizaea elegans* [9]. In the Geraniaceae family, the SSC has contracted in the genera *Viviania*, *Hypseocharis*, and *Pelargonium*, but expanded in the genera *Melianthus, Francoa,* and *California* [41]. The overall statistics revealed unique features of the *Selaginella* genus.

### 3.2. One Common and Two Independent Inversions Caused the Appearance of DR and IR Structures during Selaginella Species Evolution

Canonical plastomes are arranged in a quadripartite structure composed of an LSC and SSC region separated by two inverted repeats [37]. This is true for the plastomes of *Isoetes* and *Huperzia*. However, the *Selaginella* genus contains both inverted and direct repeats. A large inversion occurred just after the divergence of *Selaginella* from other lycophyte genera, which gave rise to DR_B_ in the *Selaginella* plastomes. DRs have also been reported in *S. tamariscina* [33], *S. vardei*, and *S. indica* [36]. Zhang et al. compared *I. flaccida* with *S. vardei*, proposing that this inversion might have occurred between 142.5 and 281.5 mya during the Late Triassic [36], which is earlier than the 64.24 mya that was estimated in this study. Nevertheless, this inversion resulted in the distinctive DRs found in the plastomes of the *Selaginella* genus, except for the three species of *S. lepidophylla*, *S. hainanensis*, and *S. uncinata*. Further inversions involving DR_B_ occurred independently in *S. lepidophylla* and in *S. hainanensis* and *S. uncinata*, in both cases restoring IR_B_ and lengthened LSCs. *S. lepidophylla* split from its sister species *S. indica* and *S. vardei* approximately 25.64 mya. Therefore, the inversion in *S. lepidophylla* must not be older than 25.64 million years old. *S. hainanensis* and *S. uncinata* diverged from each other approximately 1.03 mya, and from their sister species *S. bisulcata* and *S. pennata* approximately 4.51 mya. Therefore, this inversion must have occurred between 4.51 and 1.03 mya. The grouping of *Selaginella* species is generally in agreement with the reported phylogeny of Selaginellaceae [20,21]. However, the estimated divergence times of *Selaginella* species are relatively younger than those from previous reports [35]. These differences could be due to the fact that divergence times estimated from previous studies were obtained by utilizing a few universal genes, such as *rbcL* [20], leading to possible underestimation or overestimation. Nonetheless, the inversion events were estimated to have occurred around 246 and 23 mya [35], which is similar to the time frame our results.

IRs are an interesting feature of circular plastomes. Although one missing IR has been reported in eudicots in Fabaceae [2,42] and in some species of Geraniaceae [41], directional changes rarely occur. Other than in *Selaginella* species, only one instance of DRs has been reported, in the red alga *Porphyra purpurea* [43]. IR and DR regions contain ribosomal RNA genes and a few tRNA genes, in duplicates. Cyanobacteria and other eubacteria have one copy. It is possible that in the ancestral plastid, these genes were present in duplicates in the same orientation, but were reversed in the plastids of extant species from algae to seed plants [44]. Aside from the high demand of ribosomal RNAs for efficient translation, the IR serves to stabilize the plastome, enabling mutations to be repaired by homologous recombination between repeats through homologous sequence synapsis [37]. The synapsis of two inverted repeats can lead to the formation of a dumbbell-shaped plastid structure, as confirmed in the plastids of the common bean [37]. Alternatively, pairing between DR sequences might impose structural constraints, rather than stabilizing the circular genome. Further studies are needed to confirm the circular plastome structures of *Selaginella* species.

### 3.3. Gene and Intron Losses

Along with dynamic structural rearrangement events, *Selaginella* species showed evidence of severe gene losses that resulted in their overall small plastomes. The most noticeable losses involved *ndh* genes, encoding for NADH dehydrogenase that function in electron transfer during photosynthesis. *ndh* genes were lost or pseudogenized in several *Selaginella* species. The *ndh* genes are also absent or pseudogenized in some species of green algae and almost all gymnosperms [39]. Rhulman et al. demonstrated that the loss of plastid *ndh* genes is compensated for by the nucleus-encoded *ndh* genes in gymnosperms [45], but there are no nuclear *ndh* genes in *S. tamariscina* [33]. Zhang et al. noted that *Selaginella* species adapted to dry environments have lost their *ndh* genes, suggesting that this gene loss might be related to their adaptation to these habitats [35]. The loss of *ndh* genes was recent, as revealed by the pseudogenization of all *ndh* genes in *S. bisulcata*, whereas its sister species *S. pennata* retained the complete set: These two species only diverged 3.61 mya (Figure 3). Moreover, homologous genes to the missing ribosomal protein genes in *S. tamariscina* were found in the nuclear genome [33]. Therefore, homologous genes in the nuclear genome may have replaced the functions of some of the genes missing from the plastomes of *Selaginella* species, which could also be true for other genes.

Sixty-one codons encode 20 amino acids, but only 37 plastid tRNAs have been recognized [39]. *Selaginella* might be the genus with the most reduced number of tRNA genes in eukaryotes, as we determined in the current study. To compensate for this apparent lack of tRNA genes, Wolf et al. have proposed that post-transcriptional editing alters anticodons to code for different tRNAs [46]. However, there is no evidence for RNA editing in tRNAs from *S. uncinata* [32]. The importation of tRNAs from the nucleus has been proposed for *S. uncinata* [30] and the non-photosynthetic parasitic angiosperm *Epifagus virginiana* [47]. Alternatively, the superwobbling theory has been put forward to explain the insufficient number of plastid tRNAs [48]. This “two out of three” mechanism could allow the reading of all four nucleotides in the third codon position so that a single tRNA gene could read the corresponding codons for one four-codon family [48,49]. We also support this superwobbling theory due to the highly insufficient number of tRNAs in *Selaginella* plastomes, given the lack of clear evidence for nucleus-derived tRNA species in the plastid. 

Introns have been attracting increasing attention due to the various functions that they might have exerted during eukaryotic evolution [50]. The introns in plastid genes are self-splicing, with most being group II introns [51], except for the intron in *trnL-UAA* [52]. Nineteen plastome introns have been recognized, including 13 in protein-coding genes and 6 in tRNA genes in plastomes from green algae to seed plants [39]. Most of these introns are present in *Isoetes* and *Huperzia* species, as can be found in monilophyte ferns [6], but their numbers have been reduced by approximately half in *Selaginella* species. The deletion of intron-containing genes appears to account for the reduced number of introns in the *Selaginella* plastomes. For instance, *rps12, ycf66, rps16*, and many tRNA genes containing introns were not present in the *Selaginella* plastomes. Moreover, we observed a *Selaginella* genus-specific absence of *clpP* gene introns and the second intron of the *ycf3* gene.

### 3.4. Intraspecies Diversity

There was a wide range of intraspecies diversity within *S. tamariscina* individuals collected in different regions of Korea and China. Unlike the little variation detected among Korean collections, Chinese collections were very different at the nucleotide level (Table 2). Considering that the rate of intraspecific variation is lower in the plastome than in nuclear DNA due to the typical uniparental mode of inheritance [53], the variation detected within *S. tamariscina* individuals was very high, such as a Chinese collection—China 2019—which showed a divergence time of approximately 0.9 MYA with other *S. tamariscina* collections in China and Korea (Figure A1). Intraspecific variation may reflect the adaptation of plants to changing environmental conditions [54,55], perhaps explaining the high level of intraspecific diversity within the *S. tamariscina* collections from different regions. The morphological classification of these small primitive plants may be difficult. Therefore, with the availability of additional characteristics derived from morphological keys and molecular data, species nomenclature could become more specific.

### 3.5. A High GC Content and Abundant RNA Editing

Unlike nuclear genomes, organellar genomes are AT biased, with an average GC content of 36.45 ± 2.81% and 36.87 ± 8.23% detected in 11,542 mitochondrial genomes and plastomes, respectively [39]. Therefore, the GC contents of >50% observed in *Selaginella* plastomes are higher than those of other eukaryotic organellar genomes. AT-mutation pressure or AT-biased gene conversion (or both) is thought to have driven most plastomes to become AT biased [56,57]. Based on the nucleotide compositions of the four-fold degenerate sites in protein-coding and noncoding regions in the *S. uncinata* and *S. moellendorffii* plastomes, Smith proposed that unbiased mutation/gene conversion equilibrated the nucleotide composition to ~50% GC contents in *Selaginella* plastomes. Alternatively, he suggested that RNA editing has influenced the GC contents in these plastomes, as numerous GC-rich codons were changed into AT-rich codons through RNA editing [31].

*Selaginella* plastomes are subject to extensive RNA editing. C-to-U RNA editing is a post-transcriptional modification mechanism that grants gene diversity. There are 30–50 in plant plastomes, but *S. uncinata* contains 3415 RNA editing sites [32]. According to our results, there are numerous potential RNA editing sites in *Selaginella* plastomes. The frequency of RNA editing at the start codons of plastid genes was >50% in some *Selaginella* species. Furthermore, *S. tamariscina* RNA sequencing reads confirmed these editing sites. This high number of editing sites coincides with the exceptionally high number of pentatricopeptide repeat (PPR) protein gene families (>800) in the *S. moellendorffii* nuclear genome, representing proteins essential for plastid RNA editing [17,58]. Hecht et al. identified 2139 C-to-U RNA editing sites in the mitochondrial genome of *S. moellendorffii* [59]. Grewe et al. detected 1782 RNA editing sites in the mitochondrial genome and identified U-to-C editing in *Isoetes engelmannii* [60], indicating that the organelle genomes of lycophytes contain many RNA editing sites. If all potential sites were checked in *S. tamariscina* plastomes, the number of total events observed could be even greater than that in *S. uncinata* [32].

## 4. Materials and Methods 

### 4.1. Plant Materials and Publicly Available Plastome Sequences

One *S. tamariscina* (SNU2014) individual was collected from Baegunsan, Gwangyang-si, Jeollanam-do, Korea. A second *S. tamariscina* individual (SNU2018) was collected from Jeju Island. *S. stauntoniana* from Danyang-gun, Chungcheongbuk-do, Korea, and *S. involvens* from Jeju Island were provided by Hantaek Botanical Garden, Yongin, Gyeonggi-do, 17183, Republic of Korea.

The complete plastome sequences of 13 *Selaginella* species and three non-*Selaginella* lycophytes were obtained from NCBI (Table **1**). To analyze the intraspecies diversity in *S. tamariscina*, two plastome sequences obtained in the current study (SNU2014 and SNU2018) were used, along with assembled and annotated sequences for another Korean collection reported by Park et al. collected from Wolsong-ri, Jijeong-myeon, Wonju-si, Korea [40] and two Chinese collections reported by Xu et al. and Zhang et al. (China2018 and China2019, respectively) [33,35].

### 4.2. DNA Extraction, Sequencing, Plastome Assembly, and Annotation

Leaves were collected from the plants and ground in liquid nitrogen with a mortar and pestle. A modified cetyltrimethylammonium bromide (CTAB) method [61] was used to extract total genomic DNA from *S. tamariscina* (SNU2014), *S. stauntoniana*, and *S. involvens*. The DNA quality and concentration were assessed by gel agarose electrophoresis and UV-spectrophotometry (NanoDrop ND-1000, Thermo Fisher Scientific, Waltham, MA, USA). The extracted DNA was sequenced on the Illumina NextSeq platform, generating 7.68, 2.2, and 2.0 Gb of raw data from *S. tamariscina* (SNU2014), *S. stauntoniana*, and *S. involvens*, respectively.

The plastomes of the three species were assembled using the de novo assembly of Low-Coverage Whole genome sequence (dnaLCW) approach with the CLC genome assembler program (ver. 4.6 beta, CLC Inc., Aarhus, Denmark) [62,63]. In short, raw paired-end reads were trimmed with an offset value of 33, and trimmed reads were assembled with overlap distances set to 150 to 500 bp and the window size set to 32. Initial contigs were extracted from the assembled reads using MUMmer [64], with the reference sequence of a related species as a guide. 

For the second *S. tamariscina* individual (SNU2018), total genomic DNA was extracted using an Exgene Plant SV midi kit (GeneAll Biotechnology, Seoul, Korea). The DNA quality and concentration were assessed via agarose gel electrophoresis and UV-spectrophotometry. The extracted DNA was sequenced on a MinION sequencer from Oxford Nanopore Technologies. Library preparation was performed with a Ligation Sequencing Kit (SQK-LSK108) following the manufacturer’s instructions. One flow cell was run for 48 h. In total, 10 Gb of raw sequencing reads was filtered by removing adaptor sequences using Porechop (v0.2.0, https://github.com/rrwick/Porechop) and corrected with Canu using default parameters [65]. The Canu-corrected reads were assembled using SMARTdenovo (https://github.com/ruanjue/smartdenovo). The complete plastome sequence assembled using Illumina sequencing data was used as a reference to extract plastome-related contigs assembled by Nanopore sequencing using BLASTN.

The plastomes assembled in the previous steps were annotated using GeSeq (https://chlorobox.mpimp-golm.mpg.de/geseq.html) with complete plastome references of other species in the *Selaginella* genus and non-*Selaginella* species registered in NCBI. The annotated sequences were manually curated for details such as start and stop codons and correct coding frames using Artemis [66] For further validation of annotation, complete CDS sequences of reference sequences were collected and converted into amino acid sequences, and searched in the three assembled sequences using TBLASTN (https://blast.ncbi.nlm.nih.gov/Blast.cgi?PROGRAM=tblastn&PAGE_TYPE=BlastSearch&LINK_LOC=blasthome) for sequence homology. Furthermore, the aligned sequence was searched in the Conserved domain search provided by NCBI for complete gene structures. Genes that have sequence homology detected by TBLASTN but have incomplete gene structures due to the lack of sequences of internal stop codons were assigned as pseudogenes. Genes that were not searched at all through TBLASTN were marked as missing genes. This validation step for the pseudogene/missing gene assignment was conducted for other already published sequences used in this study.

### 4.3. Confirmation of Direct Repeat Structures by PCR Amplification

To confirm the direct repeat structure of *S. tamariscina*, primers were designed based on the junction regions between the repeat regions and the two single copies (LSC and SSC). Primers were used in different combinations to amplify the junction sites of the plastome. Hypothetical direct repeat junctions were amplified by primer combinations 1/2, 3/4, 5/2, and 3/6, while hypothetical inverted repeat junctions were amplified by different primer combinations of 1/3 and 2/4 for the first repeat block (IR_B_) (Table A1). 

PCR amplification was performed in a 25 µL volume containing 1U *Taq* polymerase, 2.5 µL of 10× reaction buffer, 0.2 mM dNTPs (Inclone Biotech, Yongin, Korea), 10 ng genomic DNA, and 10 pmol of each primer. The PCR was carried out in a thermocycler with the following parameters: 5 min at 94 °C; 28 cycles of 20 s at 94 °C, 20 s at 62 °C, and 20 s at 72 °C; and 7 min at 72 °C for the final extension step. The PCR products were visualized by electrophoresis using a 1% agarose gel with a 100 bp ladder.

### 4.4. Gene Contents and Phylogenetic Analysis

CDS, tRNA, and rRNA sequences for all 19 species of interest were extracted using FeatureExtract 1.2 L (http://www.cbs.dtu.dk/services/FeatureExtract/). Existing genes and pseudogenes were marked. Among the extracted CDS, 45 genes that were common to all 19 species were combined into one long fasta sequence for each species. These sequences were aligned using MAFFT (https://mafft.cbrc.jp/alignment/server/). The aligned sequences were trimmed using Gblocks [67] so that conserved blocks from the aligned sequences could be extracted for more accurate phylogenetic analysis. The aligned file was used for phylogenetic analysis performed using BEAST version 2.5.2. The general Time-Reversible (GTR) substitution model and a gamma category count of 4 were used for the Gamma Site Model. A random local clock and Yule tree prior were applied. A normal distribution root calibration of 375 mya representing the split between *Isoetes* and *Selaginella* families reported by Wikstrom and Kenrick [68] was used. The chain length was set to 1 million generations with a sampling frequency of 1000. The BEAST analysis was repeated four times, and the results were checked for the effective sample size (ESS) via Tracer v1.7.1. Results from all four repetitions were combined with LogCombiner implemented in the BEAST software. Moreover, a maximum clade credibility tree was drawn utilizing TreeAnnotator with a burnin percentage of 15. The final tree was visualized with FigTree v1.4.4, marking the height 95% HPD with blue node bars and node median values near the node bars. 

### 4.5. Analysis of Intron Variation and RNA Editing Sites

Total cellular RNAs were prepared from hydrated and 50% dehydrated leaf tissues with the RNeasy Plant Mini Kit (Qiagen, Hilden, Germany) by following the manufacturer’s instructions. The RNA quality and concentration were assessed using the Agilent Bioanalyzer (Agilent Technology, Santa Clara, CA, USA) and Nanodrop spectrophotometer (Thermo Fisher Scientific, Waltham, MA, USA) with parameters RIN ≥ 7, 28S:18S > 1, and OD260/280 ≥ 2.

For RNA sequencing, the cDNA libraries were constructed using TruSeq Stranded mRNA (RS-122-2101, Illumina, USA) and samples were processed with next-generation sequencing (NGS) procedures on an Illumina Hiseq3000 with a paired-end method. The NGS reads were filtered by the threshold of >20% ‘N’ bases, ≤Q20 in Phred quality, and a length > 50 bp. De novo assembly was performed using TRINITY (https://github.com/trinityrnaseq; [69]) and CD-HIT (http://weizhongli-lab.org/cd-hit/; [70]).

Genes in the plastomes of all 19 species were surveyed to check for the presence of intron regions. Potential C-to-U RNA editing sites were surveyed in the start and stop codons of the CDS from the plastomes of ten lycophyte species. Start codons with an ACG sequence instead of ATG and stop codons with CAA, CAG, and CGA sequences instead of TAA, TAG, and TGA were counted as potential RNA editing sites. Potential RNA editing sites were marked for all ten species and analyzed.

To determine whether the potential RNA editing sites had gone through the editing process, raw RNA sequencing data from *S. tamariscina*_2 were mapped onto the reference genome produced from the assembly of *S. tamariscina* (SNU2014) sequencing data. The start and stop codon positions of the CDS were obtained, and CLC find variations software (ver. 4.3.0) was used to examine the distribution of unedited mRNA reads with the C nucleotide and edited mRNA reads with the T nucleotide (previously U in RNA that changed to T during the process of RNA sequencing).

## Figures and Tables

**Figure 1 ijms-22-00641-f001:**
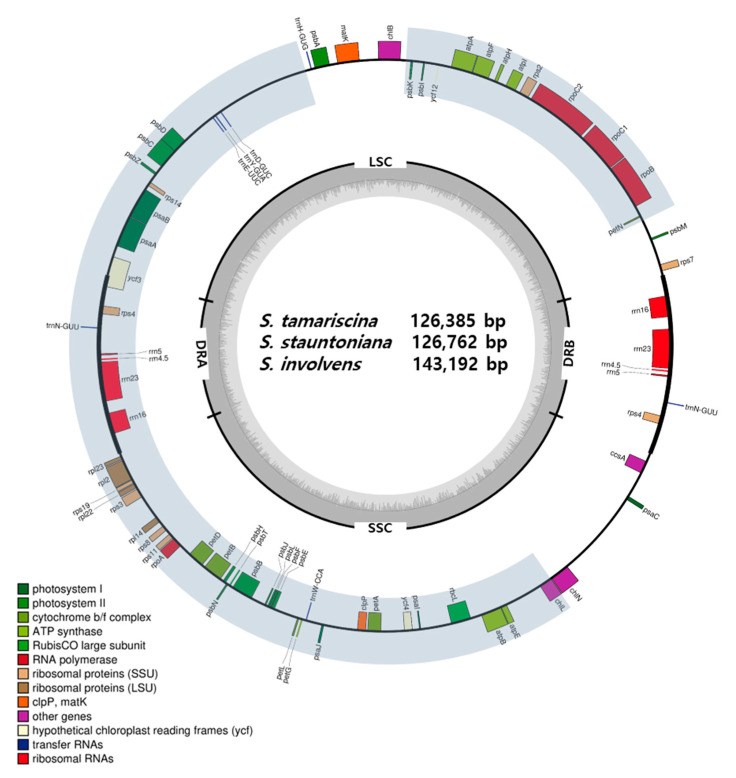
Map of complete plastid genomes of the three *Selaginella* species sequenced and assembled in this study. Shaded areas indicate regions involved in the inversion event.

**Figure 2 ijms-22-00641-f002:**
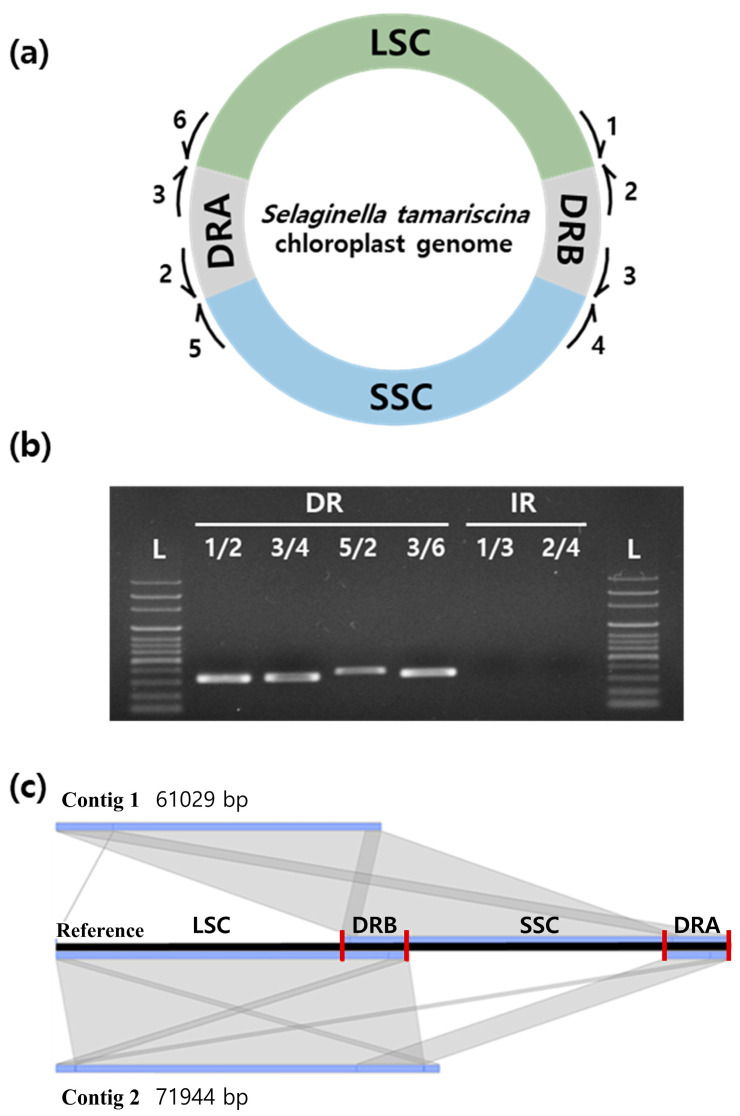
Validation of the direct repeat structures of *Selaginella tamariscina* via PCR and long-read sequencing. (**a**) Primers 1–6 were designed based on the junction regions between the single copies and repeat regions (the primer sequences are listed in Appendix A
Table A1). The primers were used in various combinations to amplify hypothetical direct repeat structures and indirect repeat structures. (**b**) 1% agarose gel electrophoresis of the PCR products. (**c**) Confirmation of the direct repeat structure in assembled contigs spanning single-copy regions and the repeat region using Oxford Nanopore Sequencing technology.

**Figure 3 ijms-22-00641-f003:**
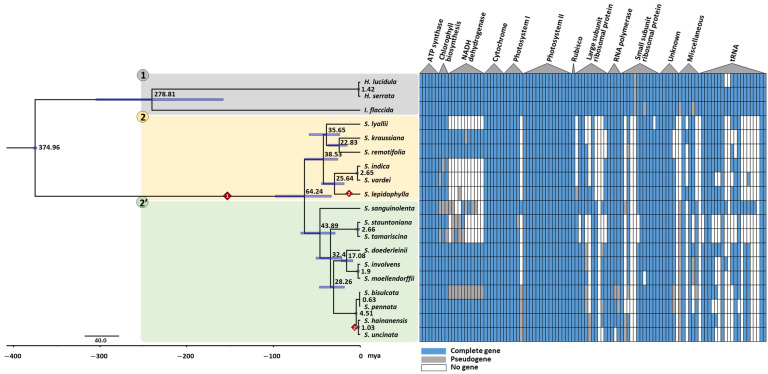
Phylogenetic analysis of the 19 lycophyte species examined in this study. The coding sequence (CDS) of 45 shared genes were used in the phylogenetic analysis with BEAST software. The calibration node was set to 375 mya for the split of *Isoetes* and *Selaginella* species. **Left**, phylogenetic tree. The gray box labeled 1 highlights non-*Selaginella* species. Lineages 2 and 2′ in the *Selaginella* genus are highlighted with yellow and green boxes, respectively. The red diamond labeled 1 represents the first inversion event, which differentiates non-*Selaginella* species from *Selaginella* species. Red diamonds labeled 2 and 2′ represent the second inversion events, which occurred independently in each lineage and converted the direct repeat structures back into inverted repeats. Blue bars on the branch of the phylogenetic tree indicate height 95% highest posterior density (HPD) and numbers near the nodes represent median values. **Right**, corresponding gene contents for each species. Blue boxes indicate genes that are present, gray boxes indicate pseudogenes, and white boxes indicate missing genes.

**Figure 4 ijms-22-00641-f004:**
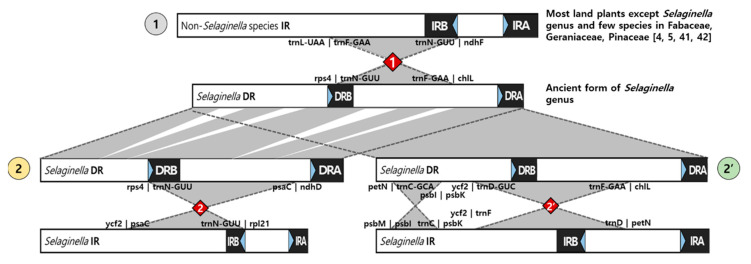
Structural rearrangements in the 19 lycophytes that diverged from non-*Selaginella* species (gray circle labeled 1) in two major lineages labeled 2 (yellow circle) and 2′ (green circle). Major inversion events containing part of the single-copy (SC) regions and one repeat region are marked with red diamonds. Each junction site is labeled with gene names to the left and right boundaries of the inversion block junction sites. All structures are drawn to scale.

**Figure 5 ijms-22-00641-f005:**
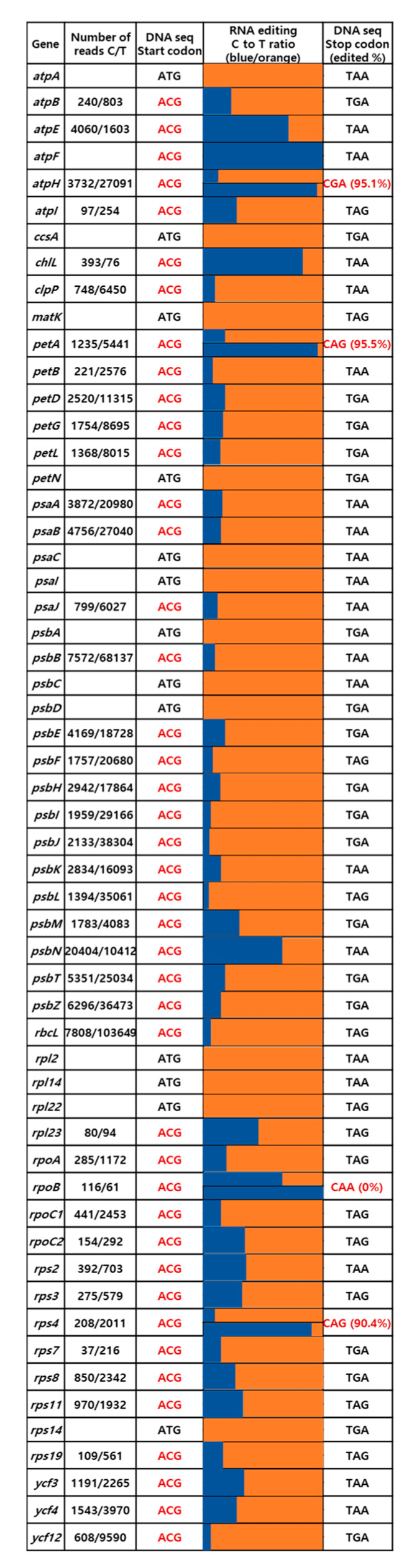
C-to-U RNA editing sites in the start and stop codons of the *S. tamariscina* chloroplast genome. RNA sequencing data from *S. tamariscina* were mapped onto the chloroplast genome sequence to identify C-to-T nucleotide distributions in the raw reads. Blue indicates reads containing C in the codon site, and orange indicates reads containing T. For genes in which both the start and stop codons were edited, the ratios of edited to non-edited reads are represented by two bars (with the bar for the start codon on top).

**Figure 6 ijms-22-00641-f006:**
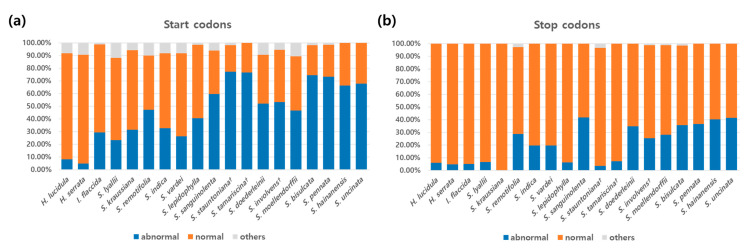
Potential C-to-U RNA editing sites in the start codons (**a**) and stop codons (**b**) of the 19 lycophyte species. Sites were checked based on the chloroplast genome sequences. Blue bars (‘abnormal’) represent the percentages of genes with the start codon ACG and stop codon CAA, CAG, or CGA, which are potential RNA editing sites. Orange bars (‘normal’) represent the percentages of genes with the normal start codon ATG and stop codons TAA, TAG, and TGA. Gray bars (‘others’) indicate the percentages of genes with start or stop codons that do not fall into either category. ^†^ Complete plastomes sequenced, assembled, and annotated in this study.

**Table 1 ijms-22-00641-t001:** Plastome information about the 19 lycophyte species examined in this study.

Order	Scientific Name	Structure Size (bp)				Gene Contents (Repeated Genes)					GC Content (%)	GenBank ID
		Total Length	LSC	SSC	IRs or DRs	Total	Protein	rRNA	tRNA	Repeat		
Lycopodiales	*Huperzia lucidula*	154,373	104,088	19,657	15,314	124	85	4(4)	27(4)	IR	36.25	NC_006861.1
	*Huperzia serrata*	154,176	104,080	19,658	15,219	130	85(2)	4(4)	30(5)	IR	36.28	NC_033874.1
Isoetales	*Isoetes flaccida*	145,303	91,862	27,205	13,118	128	82(1)	4(4)	32(5)	IR	37.94	GU191333.1
Selaginellales	*S. lyallii*	110,411	44,943	45,276	10,096	83	60(1)	4(4)	12(2)	DR	50.75	NC_041556.1
	*S. kraussiana*	129,971	46,049	54,728	14,597	92	70(3)	4(4)	10(1)	DR	52.33	NC_040926.1
	*S. remotifolia*	131,867	46,351	55,844	14,836	95	70(3)	4(4)	12(2)	DR	56.49	NC_041644.1
	*S. indica*	122,460	45,711	48,395	14,177	86	61(3)	4(4)	12(2)	DR	53.55	MK156801.1
	*S. vardei*	121,254	45,792	47,676	13,893	84	61(3)	4(4)	10(2)	DR	53.21	MG272482.1
	*S. lepidophylla*	114,693	80,625	19,452	7308	85	64	4(4)	12(1)	IR	51.94	NC_040927.1
	*S. sanguinolenta*	147,148	54,436	59,650	16,531	102	67(2)	4(4)	22(3)	DR	50.78	NC_041645.1
	*S. stauntoniana*	126,762	54,231	47,745	12,393	76	60(1)	4(4)	6(1)	DR	54.06	MK460598 ^†^
	*S. tamariscina*	126,385	53,219	47,600	12,783	75	59(1)	4(4)	6(1)	DR	53.98	MK460597 ^†^
	*S. doederleinii*	142,752	57,841	62,865	11,023	100	75(1)	4(4)	14(2)	DR	51.13	NC_041641.1
	*S. involvens*	143,192	58,193	61,075	11,962	102	75(1)	4(4)	14(4)	DR	50.82	MK460599 ^†^
	*S. moellendorffii*	143,525	58,198	61,129	12,099	99	75(1)	4(4)	12(3)	DR	51.00	MG272484.1
	*S. bisulcata*	140,509	55,598	59,659	12,626	85	59(3)	4(4)	12(3)	DR	52.77	NC_041640.1
	*S. pennata*	138,024	54,979	59,847	11,599	93	71(3)	4(4)	9(2)	DR	52.91	NC_041643.1
	*S. hainanensis*	144,201	77,780	40,819	12,801	103	77(3)	4(4)	12(3)	IR	54.83	NC_041642.1
	*S. uncinata*	144,170	77,706	40,886	12,789	101	75(3)	4(4)	11(4)	IR	54.85	AB197035.2

^†^ Complete plastomes sequenced, assembled, and annotated in this study; LSC, long-single copy; SSC, short-single copy; IR, inverted repeats; DR, direct repeats; tRNA, transfer RNA; rRNA, ribosomal RNA. Numbers in parentheses represent the number of duplicated genes due to their position in the inverted repeat region.

**Table 2 ijms-22-00641-t002:** Information about single-nucleotide polymorphisms (SNPs) and insertions/deletions (InDels) among the plastomes of the five *S. tamariscina* collections examined in this study.

		SNP				
		SNU 2014 ^†^	SNU 2018 ^†^	Korea 2020	China 2018	China 2019
**InDel**	**SNU** **2014**	**-**	**1**	**1**	**151**	**1219**
	**SNU** **2018**	**8**	**-**	**0**	**150**	**1218**
	**Korea** **2020**	**14**	**15**	**-**	**150**	**1218**
	**China** **2018**	**113**	**113**	**116**	**-**	**1246**
	**China** **2019**	**396**	**394**	**386**	**401**	**-**

Sources: SNU2014, Gwangyang-si, Jeollanam-do, Korea; SNU2018, Jeju Island, Korea; Korea2020, Wolsong-ri, Jijeong-myeon, Wonju-si, Korea, [40]; China2018, [32]; China2019, [34]. ^†^ Complete plastomes sequenced, assembled, and annotated in this study. The gray gradient color scheme displays the increasing SNP/InDel numbers.

**Table 3 ijms-22-00641-t003:** Presence and absence of introns in intron-containing genes in the 19 lycophyte species.

Function	Gene	*H. lu*	*H. se*	*I. fl*	*S. ly*	*S. kra*	*S. rem*	*S. ind*	*S. va*	*S. lep*	*S. san*	*S. st* ^†^	*S. ta* ^†^	*S. doe*	*S. inv* ^†^	*S. mo*	*S. bis*	*S. pen*	*S. hai*	*S. un*
**ATP synthase**	***atpF***	**+**	**+**	**+**	**+**	**+**	**+**	–	–	**+**	**+**	**+**	**+**	**+**	**+**	**+**	**+**	**+**	**+**	**+**
**Other**	***clpP-1***	**+**	**+**	**+**	–	–	–	–	–	–	+	–	–	**+**	**+**	**+**	**+**	**+**	**+**	**+**
	***clpP-2***	**+**	**+**	**+**	–	–	–	–	–	–	**+**	–	–	**+**	–	–	–	–	–	–
**NADH dehydrogenase**	***ndhA***	**+**	**+**	**+**		**+**	**+**							**+**	**+**	**+**		**+**	**+**	**+**
	***ndhB***	**+**	**+**	**+**		**+**	**+**							**+**	**+**	**+**		**+**	**+**	**+**
**Cytochrome**	***petB***	**+**	**+**	**+**	**+**	**+**	**+**	**+**	**+**	**+**	**+**	**+**	**+**	**+**	**+**	**+**	**+**	**+**	**+**	**+**
	***petD***	**+**	**+**	**+**	**+**	**+**	**+**	**+**	**+**	**+**	**+**	**+**	**+**	**+**	**+**	**+**	**+**	**+**	**+**	**+**
**Large subunit ribosomal protein**	***rpl16***	**+**	**+**	**+**	**+**	**+**	**+**	**+**	**+**	**+**	**+**			**+**	**+**	**+**	**+**	**+**	**+**	**+**
	***rpl2***	**+**	**+**	**+**	**+**	**+**	**+**	**+**	**+**	**+**	**+**	**+**	**+**	**+**	**+**	**+**	**+**	**+**	**+**	**+**
**RNA polymerase**	***rpoC1***	**+**	**+**	**+**	–	**+**	**+**	**+**	**+**	**+**	**+**	**+**	**+**	**+**	**+**	**+**		**+**	**+**	**+**
**Small subunit ribosomal protein**	***rps12***	**+**	**+**	**+**																
	***rps16***	**+**	**+**	–																
**Unknown**	***ycf3–1***	**+**	**+**	**+**	+	**+**	**+**	**+**	**+**	**+**	**+**	**+**	**+**	**+**	**+**	**+**	**+**	**+**	**+**	**+**
	***ycf3–2***	**+**	**+**	**+**	–	–	–	–	–	–	**+**	**+**	**+**	**+**	**+**	**+**	**+**	**+**	**+**	**+**
	***ycf66***	–	**+**	**+**																
**tRNA**	***trnL***	**+**	**+**	**+**	–		–			–	–			–						
	***trnG***	–	–	**+**		–				–	–			–						
	***trnI***		**+**	**+**							–				–	–				
	***trnA***	**+**	**+**	**+**																
	***trnK***		**+**	**+**																
	***trnV***	**+**	**+**	**+**																

^†^ Complete plastomes sequenced, assembled, and annotated in this study; ‘+’ indicates intron-containing genes; ‘−‘ indicates genes without introns; gray boxes indicate missing genes or pseudogenes; *H. lu*, *Huperzia lucidula*; *H. se*, *Huperzia serrata*; *I. fl*, *Isoetes flaccida*; *S. ly*, *Selaginella lyallii*; *S. kra*, *S. kraussiana*; *S. rem*, *S. remotifolia*; *S. ind*, *S. indica*; *S. va*, *S. vardei*; *S. lep*, *S. lepidophylla*; *S. san*, *S. sanguinolenta*; *S. st*, *S. stauntoniana*; *S. ta*, *S. tamariscina*; *S. doe*, *S. doederleinii*; *S. inv*, *S. involvens*; *S. mo*, *S. moellendorffii*; *S. bis*, *S. bisulcata*; *S. pen*, *S. pennata*; *S. hai*, *S. hainanensis*; and *S. un*, *S. uncinata.*

## Data Availability

The data presented in this study are openly available in the National Center for Biotechnology Information under the accession numbers presented in Table 1.

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
