# Peer review of "Plastid Genomes of the Early Vascular Plant Genus Selaginella Have Unusual Direct Repeat Structures and Drastically Reduced Gene Numbers"

_ijms, 2021, doi:10.3390/ijms22020641_

Round 1

Reviewer 1 Report

The authors have sequenced and assembled the plastid genome of some Selaginella species. Comparative analyses of 19 lycophytes including two 24 Huperzia and one Isoetes were applied. Moreover, other characteristics including small genome size, drastic reductions in gene and intron numbers, high GC content, and extensive RNA editing are presented in this study.

Although the data presented are interesting and rich of important information, the manuscript show lack of clarity in presentation. The results need to be presented in a more clear manner. The organization of the Results section is an important step of the manuscript. The authors should announce systematically and clearly the study findings and should avoid to include interpretations or comparisons with literature published. In some cases, it seems that the Results section is a mixture between the results obtained and the discussion. Moreover, the authors tend to repeat or report the results in the Discussion. This mode distracts the reader from the more important points of the discussion. Thus, I suggest a complete rewrite of the sections Results and Discussion.

I have also some methodological questions:

Line 496. A strict clock was used in divergent times estimation. I recommend to avoid this type of clock because it is unrealistic in a phylogeny of 375 Mya; probably, a relaxed or local clock is more appropriate.  Generally, relaxed molecular clock models appear to fit real data better than a strict molecular clock. However, it is important to explore how clock-like your data is. For example, you can perform both a strict clock and relaxed clock analysis and let the data decide which one is most appropriate. Moreover, I suggest to incorporate multiple calibration points (that may not be consistent with a strict molecular clock) in your analysis and these calibration points should be associated with the internal nodes of the tree (see Drummond and Bouckaert ,Bayesian evolutionary analysis with BEAST 2, the book).

Line 499. All Selaginella species were grouped into a monophyletic group. Why? This procedure could influence the topology of phylogenetic tree. You must rerun the analysis without this constrain. Moreover, you have omitted the number of runs that you have produced. You should produce 4-5 runs and then to combine the results (for example using logcombiner, implemented in Beast). A visual inspection of single runs, using the Tracer software, could be sufficient to verify if you have sufficient mix.

Line 206. The paragraph 2.4 is very interesting but is not clear for me how plastome diversity among five S. tamariscina collections reflects geographical diversity. I suggest to validate the hypothesis with a specific test.

Line 167. In figure 3, you show the pseudogenes in the gray boxes. Can you explain how you find the pseudogenes in your dataset. Which software have you used? Which is the procedure that you have applied?

Author Response

Responses to Reviewer 1

The authors have sequenced and assembled the plastid genome of some Selaginella species. Comparative analyses of 19 lycophytes including two 24 Huperzia and one Isoetes were applied. Moreover, other characteristics including small genome size, drastic reductions in gene and intron numbers, high GC content, and extensive RNA editing are presented in this study.

Although the data presented are interesting and rich of important information, the manuscript show lack of clarity in presentation. The results need to be presented in a more clear manner. The organization of the Results section is an important step of the manuscript. The authors should announce systematically and clearly the study findings and should avoid to include interpretations or comparisons with literature published. In some cases, it seems that the Results section is a mixture between the results obtained and the discussion. Moreover, the authors tend to repeat or report the results in the Discussion. This mode distracts the reader from the more important points of the discussion. Thus, I suggest a complete rewrite of the sections Results and Discussion.

  • Thank you for taking your time to review and give feedback on the manuscript. I appreciate your comments and advice and tried my best to incorporate your suggestions in the analysis and answer your questions. Also, I have revised the result and discussion sections according to your advice so that the results section only contain results and the discussion section does not have repeated context from the result section. I hope the distinction is clearer. Below are the answers to your methodological questions. The line reference is based on the manuscript with “Track changes” on.

I have also some methodological questions:

Line 496. A strict clock was used in divergent times estimation. I recommend to avoid this type of clock because it is unrealistic in a phylogeny of 375 Mya; probably, a relaxed or local clock is more appropriate.  Generally, relaxed molecular clock models appear to fit real data better than a strict molecular clock. However, it is important to explore how clock-like your data is. For example, you can perform both a strict clock and relaxed clock analysis and let the data decide which one is most appropriate. Moreover, I suggest to incorporate multiple calibration points (that may not be consistent with a strict molecular clock) in your analysis and these calibration points should be associated with the internal nodes of the tree (see Drummond and Bouckaert ,Bayesian evolutionary analysis with BEAST 2, the book).

  • Thank you for the kind and detailed advice on the phylogenetic analysis. As you suggested, I performed divergence time estimations using relaxed, local, and strict clocks, without the monophyletic grouping of all Selaginella species, each with 5 runs combined with logcombiner. All clock models showed results that grouped Selaginella species together but with slightly different subgrouping. Among the clock models, the strict clock seemed to be most fitting for two reasons:
  • One, as early vascular plants, Selaginella species are known to have proliferated in the Carboniferous era (359-299 mya) according to fossil evidence of ancestor species and DNA sequence-based divergence time estimations. Divergence times estimated by the other clock models show divergence times in later geological periods which should be periods of proliferation of angiosperms.
  • Two, the divergence times estimated by the strict clock model agrees with other divergence times estimated by previously published researches.
  • About calibration points, we were only able to use one calibration point (375mya) because this is the only one that was available from fossil and DNA sequence-based analysis.
  • The newly analyzed phylogenetic tree with the above methods was updated in figure 3.

Line 499. All Selaginella species were grouped into a monophyletic group. Why? This procedure could influence the topology of phylogenetic tree. You must rerun the analysis without this constrain. Moreover, you have omitted the number of runs that you have produced. You should produce 4-5 runs and then to combine the results (for example using logcombiner, implemented in Beast). A visual inspection of single runs, using the Tracer software, could be sufficient to verify if you have sufficient mix.

  • For the BEAST analysis, I removed the grouping option of all Selaginella species into a monophyletic group. Also, I produced 5 runs for each clock model and combined the results using logcombiner provided by BEAST. I have updated the methods section on changes made to the divergence time estimation analysis (lines 581-582).

Line 206. The paragraph 2.4 is very interesting but is not clear for me how plastome diversity among five S. tamariscina collections reflects geographical diversity. I suggest to validate the hypothesis with a specific test.

  • In the intraspecies diversity analysis, we were able to use the sequences that were available in the public database and the ones assembled in our study. Through the analysis, we wanted to show that there could be many variations even within one species which could provide taxonomic clues for further sub-speciation. Although what you have suggested would be an interesting topic, I think it is difficult to validate the hypothesis in this study, especially in the current situation. These are my thoughts on geographic diversity. As you can see from table 2, individuals in Korea had very little variation even though they were collected from three different locations far from each other. The Korean peninsula is small, and the habitats of Selaginella species are usually similar (in shaded areas near rocks) so there may not have been diversification going on. I was not able to obtain geographical information of collections from China, but since Selaginella species inhabited the Earth for long periods of time, the two Chinese collections could have been from completely different environments proliferating in isolation.

Line 167. In figure 3, you show the pseudogenes in the gray boxes. Can you explain how you find the pseudogenes in your dataset. Which software have you used? Which is the procedure that you have applied?

  • I apologize for the lack of information on the methodology regarding pseudogenes. For sequences assembled in our study, we annotated them using GeSeq (https://chlorobox.mpimp-golm.mpg.de/geseq.html) and curated them using the Artemis software by manually checking start/stop codons and proper frames. As long as I know of, there isn’t a software for the detection of pseudogenes because the gene annotation process itself has to go through manual curation because the programs cannot detect proper frames or start/stop codons 100% of the time. The sequence information from our study as well as those registered in NCBI were double checked to identify pseudogenes by first collecting complete CDS sequences of from the reference sequences as the database and searching for the gene using tblastn, and further confirming the structure using Conserved domain search provided from NCBI. I have added the procedure in the material and method section (lines 547-556).

Reviewer 2 Report

The purpose of this paper was to sequence and analyze the features of the chloroplast genome of Selaginella species, a group of early land plants with unique plastic genome features.  The authors performed DNA extraction and analysis for two S. tamariscina individuals, one S. stauntoniana, and one S. involvens and compared these data to publicly available sequence data for other lycophyte plastome sequences.  They found that their specimens exhibited many of the known features of other Selaginella species, such as high GC content, small plastome sizes, and the presence of both inverted and direct repeats.  These samples also showed high levels of C to U base changes, which was a bit confounding for identification of start codons. 

A few comments

  1. It would be very helpful to have a phylogeny early on in the figures illustrating the relationships between the species and the types of repeats found in their plastomes. Ideally this figure would show the main clades of land plants and their key plastome features. 

  1. The discussion mentions the topic of genes that are missing from the plastome (or turned into pseudogenes) are often present in the nuclear genome. More detail on the tRNA encoded genes would be useful.  Does is appear that the “super wobbling” theory would fit with the pattern of which tRNAs are absent?  It unclear how a wobble in the third position would allow for coding of 10 different amino acids (seems that 4 would be the limit, given that there are 4 bases for that position). 

  1. How much of a concern is sequence quality or specimen identity when looking for SNPs between the individuals shown in Table 2? It seems that uneven quality of data could give an artificially high number of SNPs.  As mentioned in the text, it is possible that one of these samples belongs to a different species.  How was species identification determined for these samples? 

  1. Is there evidence for plastome diversity in a single organism (as there is for mitochondria)? Given that plastids are generally inherited from a single parent any within individual variation could be a source for diversity. 

  1. It would be helpful to include images of the species studied, ideally including microscopy of the chloroplasts. Do these Selaginella species have chloroplast features which are different from the Lycopodiales or Isotales such as size, abundance, distribution?  Or are the habitats different?  Some context for the chloroplasts in reference to the biology of the organisms would be helpful. 

Minor comments

  1. It would help to highlight which data are from this study and which is from databases (such as in Table 1).

  1. Please add a citation for lines 36-38 regarding general plastic genome structure.

  1. As currently shown, the text in figure 1 is very small, please enlarge it.

  1. What are the numbers in parenthesis in Table 1? Please clarify. 

  1. What are the grey shaded areas of Table 3? Missing genes?  Please clarify

  1. The blue/orange coloration of Figure 5 does not print well at all in greyscale, please consider different shades. Also, rather than having the genes in order by name, consider ordering them by RNA editing levels (most to least). 

  1. Consider re-ordering the bars in Figure 6 by % of codons in each category.

Author Response

Responses to Reviewer 2

The purpose of this paper was to sequence and analyze the features of the chloroplast genome of Selaginella species, a group of early land plants with unique plastic genome features.  The authors performed DNA extraction and analysis for two S. tamariscina individuals, one S. stauntoniana, and one S. involvens and compared these data to publicly available sequence data for other lycophyte plastome sequences.  They found that their specimens exhibited many of the known features of other Selaginella species, such as high GC content, small plastome sizes, and the presence of both inverted and direct repeats.  These samples also showed high levels of C to U base changes, which was a bit confounding for identification of start codons. 

  • Thank you for taking your time to review and give feedback on the manuscript. I appreciate your comments and advice and tried my best to incorporate your suggestions in the manuscript and answer your questions. Here are the answers to your question below. The line reference is based on the manuscript with “Track changes” on.

A few comments

1.It would be very helpful to have a phylogeny early on in the figures illustrating the relationships between the species and the types of repeats found in their plastomes. Ideally this figure would show the main clades of land plants and their key plastome features. 

  • As mentioned in lines 36-38, a quadripartite organization of the plastomes is the typical structure found in all land plants. The loss of one of the repeat pair was reported in some conifers and some Fabaceae species (lines 78-82), but there is no land plant reported of up to date on the direct repeat structure like those found in the genus Selaginella. For this reason, I didn’t find it necessary to show a figure that shows the phylogeny of land plants with their repeat structures when they’d all be inverted structures. Instead, I edited figure 4 by adding a phrase next to the plastome structure of lineage 1 so that this plastome structure is typical in almost all land plant species.

2. The discussion mentions the topic of genes that are missing from the plastome (or turned into pseudogenes) are often present in the nuclear genome. More detail on the tRNA encoded genes would be useful.  Does is appear that the “super wobbling” theory would fit with the pattern of which tRNAs are absent?  It unclear how a wobble in the third position would allow for coding of 10 different amino acids (seems that 4 would be the limit, given that there are 4 bases for that position). 

  • I think there has been a mistake in the delivery of the sentence about the superwobbling theory allowing one tRNA gene to be able to read ten amino acids. You are right about the fact that 4 is the limit. I have made changes from “for 10 different amino acids” to “for one four-codon family” in line 441.

3. How much of a concern is sequence quality or specimen identity when looking for SNPs between the individuals shown in Table 2? It seems that uneven quality of data could give an artificially high number of SNPs.  As mentioned in the text, it is possible that one of these samples belongs to a different species.  How was species identification determined for these samples? 

  • Yes, if the quality of the sequence is bad, the results could be skewed. However, sequencing and assembly techniques have improved so much that this is of little to almost no concern. In our lab, we have sequenced, assembled, and annotated hundreds of plastome sequences in the past 10 years and developed a pipeline for plastome sequence assembly (dnaLCW mentioned in line 527). Furthermore, we were able to develop high-quality sequence-based markers to distinguish and identify samples. We could say from experience that the assembly with little to no error is possible for the plastome sequence. This would not be possible with bad quality sequence information. The sequence information of S. tamariscina individuals in table 2 were relatively recently completed with third generation (PacBio) and next generation sequencing technologies (Illumina) from which we could assume that the complete plastome sequences would have very little to no error, few enough to not affect the analysis to the point where it would contribute to artificially high number of SNPs. As for specimen identity, usually experts determine the species by morphological traits which was done for samples from our study. Further identification could be done with molecular based markers from plastomes, but usually different species display a different scale of variations. What was previously written in the results section (was in line 246, but now removed) was worded in a way that could cause misunderstanding. Here, we wanted to show that even within one species, there is a high number of variations which could be due to the fact that they have been inhabiting the Earth for long periods of time proliferating and accumulating variations in their respective habitats. Morphological, and taxonomic studies are still being done on this species, and these variations could possibly mean that further sub-speciation could be done. Nonetheless, for now, China2019 is classified as a S. tamariscina individual since it is still far away from the closest sister species, S. stauntoniana (figure A1), which means that there is even more variation than those analyzed between Chinese collections.

4. Is there evidence for plastome diversity in a single organism (as there is for mitochondria)? Given that plastids are generally inherited from a single parent any within individual variation could be a source for diversity. 

  • There are some cases of plastome diversity derived from heteroplasmy in evening primroses (Oenothera) where the plastome is biparentally inherited. However, this is one of the few and rare cases of heteroplasmy in a single organism, and the rest show uniparentally inherited plastomes. Therefore, most plants have homoplasmy in one organism. However, there is some range of plastome nucleotide diversity among population in one species.

5. It would be helpful to include images of the species studied, ideally including microscopy of the chloroplasts. Do these Selaginella species have chloroplast features which are different from the Lycopodiales or Isotales such as size, abundance, distribution?  Or are the habitats different?  Some context for the chloroplasts in reference to the biology of the organisms would be helpful.

  • Unfortunately, I do not have pictures of the species studied nor the microscopic pictures to display in the manuscript since collecting and sequencing was done a few years back. However, recent studies on morphology of chloroplasts in relation to other species such as early endosymbiont species show that early vascular plants including Lycopodiales or Isoetales have a mixture of chloroplast structures such as size and abundance. Even within the same Selaginella species, monoplastidic and polyplastidic chloroplast exists in different sizes and different habitats.

Minor comments

6. It would help to highlight which data are from this study and which is from databases (such as in Table 1).

  • I have added the footnotes and phrases indicating which data are from this study in table 2, table 3, and figure 6, just like I did in table 1.

7. Please add a citation for lines 36-38 regarding general plastic genome structure.

  • I have added the citation for the typical quadripartite plastid genome structure in line 38.

8. As currently shown, the text in figure 1 is very small, please enlarge it.

  • I enlarged figure 1 to fit the size of the paper. I hope the figure/text is more visible now.

9. What are the numbers in parenthesis in Table 1? Please clarify. 

  • Numbers in parentheses represent the number of genes in duplicates because they are positioned in the inverted repeat pair. I added the explanation in line 117.

10. What are the grey shaded areas of Table 3? Missing genes?  Please clarify

  • The grey shaded areas indicate missing genes (or pseudogenes). I am sorry for omitting detailed explanation of signs/shades with the table. I included the explanations on line 275.

11. The blue/orange coloration of Figure 5 does not print well at all in greyscale, please consider different shades. Also, rather than having the genes in order by name, consider ordering them by RNA editing levels (most to least). 

  • I have changed the shades so that they could be distinguishable in greyscale. About the ordering of RNA editing levels in the order of most to least, I think it would be better to leave it in alphabetical order because this way we can see how the genes are grouped together and see the pattern or trend of RNA editing more abundant in specific gene families or genes of similar function.

12. Consider re-ordering the bars in Figure 6 by % of codons in each category.

  • The bars were ordered based on the species ordering in the phylogenetic tree of figure 3. The purpose of this graph was to show that the Selaginella genus had high frequencies of potential RNA editing sites, but without a specific trend or pattern. By changing the order according to percentages will reposition non-Selaginella species within Selaginella species and will be in a completely different order for start and stop codons. I worry that these could confuse the original message of the figure.

Round 2

Reviewer 1 Report

The manuscript appears to have been significantly improved. However, the evolutionary model is a fundamental aspect phylogeny not fully explained in the manuscript.  For example,I have recommend to avoid the strict clock because it is unrealistic in a phylogeny of 375 Mya.
I disagree with the justifications of the authors and I think that a specific test to verify the best evolutionary model should be applied. In recent years, several new approaches are been developed to perform model selection in the field of phylogenetics, such as path sampling, stepping-stone sampling and generalized stepping-stone sampling (https://beast.community/model_selection_1). As reported in my previous revision, the strict clock does not seem plausible (often used a-priori in intra-species  phylogeny); perhaps more likely the random local clock. I suggest that the authors read carefully the problems that can exist if the best model is not selected (Baele et al., 2012, 2013 and 2016).
The authors should not try to evade the problem but to address it using a specific method. Methods and results must be well described and detailed in the main text.

Moreover, I have some doubts about the phylogenetic tree in Figure A1. Little information is described on how the analysis was performed and more details should be reported in the text. In this case, a strict clock may be adequate but I would not use a speciation model (e.g. yule). The plastomes are all accessions of the same species, so a coalescence model should be more adequate. You should describe in detail how the analyzes were done (I believe you have used a secondary calibration) and which prior have you used. I suggest to read HO AND PHILLIPS 2009, Ho et al., 2005; Ho et al., 2011 on these topics.

In Figure 3 and Figure A1 are not reported the 95%HPD interval for node ages

G. Baele, P. Lemey, T. Bedford, A. Rambaut, M. A. Suchard, M. A. and A. V. Alekseyenko (2012) Improving the accuracy of demographic and molecular clock model comparison while accommodating phylogenetic uncertainty. Mol. Biol. Evol. 29 (9), 2157-2167.

G. Baele, W. L. S. Li, A. J. Drummond, M. A. Suchard, and P. Lemey (2013) Accurate model selection of relaxed molecular clocks in Bayesian phylogenetics. Mol. Biol. Evol. 30 (2): 239-243.

G. Baele, P. Lemey and M. A. Suchard (2016) Genealogical working distributions for Bayesian model testing with phylogenetic uncertainty. Syst. Biol. 65(2), 250-264.
N. Lartillot and H. Philippe (2006) Computing Bayes factors using thermodynamic integration. Syst. Biol. 55:195–207.

S. Y. W. HO AND M. J. PHILLIPS 2009. Accounting for Calibration Uncertainty in Phylogenetic Estimation of Evolutionary Divergence Times Syst. Biol. 58(3):367–380,

S. Y. W. HO et al., 2005. Time Dependency of Molecular Rate Estimates and Systematic Overestimation of Recent Divergence Times Mol. Biol. Evol. 22(7):1561–1568.

S. Y. W. HO et al., 2011.Time-dependent rates of molecular evolution Mol. Ecol.

Author Response

Reviewer’s comments

The manuscript appears to have been significantly improved. However, the evolutionary model is a fundamental aspect phylogeny not fully explained in the manuscript.  For example,I have recommend to avoid the strict clock because it is unrealistic in a phylogeny of 375 Mya.
I disagree with the justifications of the authors and I think that a specific test to verify the best evolutionary model should be applied. In recent years, several new approaches are been developed to perform model selection in the field of phylogenetics, such as path sampling, stepping-stone sampling and generalized stepping-stone sampling (https://beast.community/model_selection_1). As reported in my previous revision, the strict clock does not seem plausible (often used a-priori in intra-species  phylogeny); perhaps more likely the random local clock. I suggest that the authors read carefully the problems that can exist if the best model is not selected (Baele et al., 2012, 2013 and 2016).
The authors should not try to evade the problem but to address it using a specific method. Methods and results must be well described and detailed in the main text.

Moreover, I have some doubts about the phylogenetic tree in Figure A1. Little information is described on how the analysis was performed and more details should be reported in the text. In this case, a strict clock may be adequate but I would not use a speciation model (e.g. yule). The plastomes are all accessions of the same species, so a coalescence model should be more adequate. You should describe in detail how the analyzes were done (I believe you have used a secondary calibration) and which prior have you used. I suggest to read HO AND PHILLIPS 2009, Ho et al., 2005; Ho et al., 2011 on these topics.

In Figure 3 and Figure A1 are not reported the 95%HPD interval for node ages

Baele, P. Lemey, T. Bedford, A. Rambaut, M. A. Suchard, M. A. and A. V. Alekseyenko (2012) Improving the accuracy of demographic and molecular clock model comparison while accommodating phylogenetic uncertainty. Mol. Biol. Evol. 29 (9), 2157-2167.

Baele, W. L. S. Li, A. J. Drummond, M. A. Suchard, and P. Lemey (2013) Accurate model selection of relaxed molecular clocks in Bayesian phylogenetics. Mol. Biol. Evol. 30 (2): 239-243.

Baele, P. Lemey and M. A. Suchard (2016) Genealogical working distributions for Bayesian model testing with phylogenetic uncertainty. Syst. Biol. 65(2), 250-264.
N. Lartillot and H. Philippe (2006) Computing Bayes factors using thermodynamic integration. Syst. Biol. 55:195–207.

Y. W. HO AND M. J. PHILLIPS 2009. Accounting for Calibration Uncertainty in Phylogenetic Estimation of Evolutionary Divergence Times Syst. Biol. 58(3):367–380,

Y. W. HO et al., 2005. Time Dependency of Molecular Rate Estimates and Systematic Overestimation of Recent Divergence Times Mol. Biol. Evol. 22(7):1561–1568.

Y. W. HO et al., 2011.Time-dependent rates of molecular evolution Mol. Ecol.

Response to Reviewer’s Comment

Dear Reviewer,

Thank you for your valuable and detailed feedback. We were quite unfamiliar with the details that had to go into the analyses, and your feedback provided very nice guidelines for a deeper understanding of the phylogenetic analysis. We tried our best to incorporate your advice to improve the quality of our data. Through reanalysis, we were able to conclude that the strict clock that we had previously used was inappropriate for the phylogeny of 375 mya, especially when the strict clock is normally used for analyzing populations within the same species. Results from other clocks (relaxed) disagreed with previously reported and well-known taxonomical classification of Selaginella species, causing some of them to be grouped with non-Selaginella species. The results produced by the random clock was most plausible for correct classification. Although the divergence times were estimated to be lower (closer to 0 mya on the evolutionary scale) than our previous results and other previously reported phylogenies, it has to be taken into consideration that: one, these species as well as their extinct or extant ancestors have been around for a long time inhabiting various regions worldwide being highly diverse; and two, previous phylogenetic research has been conducted by using only a few universal genes such as rbcL without fossil data, which can lead to under/overestimation of divergence times.

For the phylogenetic tree of Figure A1, we reanalyzed the data using the strict clock and the coalescence constant population model as you have suggested. And yes, a secondary calibration derived from the main phylogenetic analysis with the 19 lycophyte species was used. Here, plastome sequences of 5 S. tamariscina individuals and 1 sister species, S. stauntoniana, were compared. The divergence event of these two species was estimated to be around 2.66 mya (relatively recent), but we do not think that exact estimations separating individuals within the S. tamariscina population is as critical to the main flow of the text. This analysis was done to support the highly variable intraspecies diversity within the S. tamariscina population, which could suggest the diversity within the Selaginella genus itself.

Before making changes for the second revision, I accepted/applied all changes from the previous revision before turning ‘Track changes’ on to make revisions made this round more visible. I hope this is okay.

Figures 3 and A1 were replaced with the new analyzed figures, and the 95% HPD interval was added on as blue bars. Methodological details discussed above have been added in the manuscript (lines 568-577 for Figure 3, lines 625-633 for Figure A1, with ‘Track changes’ on). We have also added a few lines regarding the different divergence time estimations compared to those of previous reports (lines 361-366). We also made some minor grammar/format changes throughout the text.

If there is more to be worked or improved on, please do not hesitate to let us know. We really appreciate your feedback.

Sincerely,

Hyeonah Shim

Round 3

Reviewer 1 Report

For me is ok